# Porcine UL-16 Binding Protein 1 Is Not a Functional Ligand for the Human Natural Killer Cell Activating Receptor NKG2D

**DOI:** 10.3390/cells12222587

**Published:** 2023-11-07

**Authors:** Kevin J. Lopez, John Paul Spence, Wei Li, Wenjun Zhang, Barry Wei, Arthur A. Cross-Najafi, James R. Butler, David K. C. Cooper, Burcin Ekser, Ping Li

**Affiliations:** 1Department of Surgery, Indiana University School of Medicine, Indianapolis, IN 46202, USA; kevlopez@iu.edu (K.J.L.); wenzhang@iu.edu (W.Z.); barrwei@iu.edu (B.W.); acrossna@iupui.edu (A.A.C.-N.); jrbutler@iupui.edu (J.R.B.); 2Department of Pediatrics, Indiana University School of Medicine, Indianapolis, IN 46202, USA; jspence@iupui.edu; 3Department of Microbiology and Immunology, Indiana University School of Medicine, Indianapolis, IN 46202, USA; wl1@iupui.edu; 4Center for Transplantation Sciences, Massachusetts General Hospital, Harvard Medical School, Charlestown, MA 02129, USA; dkcooper@mgh.harvard.edu

**Keywords:** xenotransplantation, natural killer (NK) cell, ULBP-1, NKG2D, immune rejection, gene editing

## Abstract

Natural killer (NK) cells play a vital role in xenotransplantation rejection. One approach to induce NK cell immune tolerance is to prevent the NK cell-mediated direct killing of porcine cells by targeting the interaction of the activating receptor NKG2D and its ligands. However, the identity of porcine ligands for the human NKG2D receptor has remained elusive. Previous studies on porcine UL-16 binding protein 1 (pULBP-1) as a ligand for human NKG2D have yielded contradictory results. The goal of the present study was to clarify the role of pULBP-1 in the immune response and its interaction with human NKG2D receptor. To accomplish this, the CRISPR/Cas9 gene editing tool was employed to disrupt the porcine *ULBP-1* gene in a 5-gene knockout porcine endothelial cell line (*GGTA1*, *CMAH*, *β4galNT2*, *SLA-I α chain*, and *β-2 microglobulin*, 5GKO). A colony with two allele mutations in *pULBP*-1 was established as a 6-gene knockout pig cell line (6GKO). We found that pULBP-1-deficient pig cells exhibited a reduced binding capacity to human NKG2D-Fc, a recombinant chimera protein. However, the removal of ULBP-1 from porcine endothelial cells did not significantly impact human NK cell degranulation or cytotoxicity upon stimulation with the pig cells. These findings conclusively demonstrate that pULBP-1 is not a crucial ligand for initiating xenogeneic human NK cell activation.

## 1. Introduction

Xenotransplantation offers a promising solution to the critical shortage of human organs for transplantation. Previous studies suggest that the transplantation of pig organs into humans might provide a viable option for this purpose [1]. Indeed, recent progress in pig genetic engineering has significantly advanced the potential for incorporating xenotransplantation into clinical settings [2]. This approach involves the removal of xenoantigens from vital organs as well as introducing human immune regulatory proteins to these organs to limit the deleterious immune response. For instance, the transplantation of a genetically modified pig heart into a human supported a patient for two months, thereby marking a milestone in pig-to-human xenotransplantation in practice. Therefore, the prevention or elimination of hyperacute immune rejection is feasible [3]. However, further investigations revealed endothelial injuries in the xenograft, potentially resulting from antibody-mediated rejection [4]. Thus, the persistent challenge of immune rejection remains a significant hurdle due to the inherent immune-related incompatibilities between humans and pigs. Identifying the specific mechanisms that aim at extending the survival of xenografts is necessary to improve the viability of pig-to-human xenotransplantation.

Previous studies identified the infiltration and activation of natural killer (NK) cells in both allotransplantation and xenotransplantation [5,6]. NK cells are vital components of the innate immune system that play a central role in recognizing and eliminating non-self or stressed cells [7]. NK cells lead to the immune rejection of transplanted organs due to specific mechanisms which attack the endothelium. For instance, antibody-dependent cell-mediated cytotoxicity (ADCC) acts via FcγRIIIA (CD16) binding to the Fc region of human IgG on porcine endothelium [8]. In addition, direct cytotoxicity is mediated through receptor–ligand interactions [9,10]. Lastly, the secretion of pro-inflammatory molecules like interferon-γ (IFN-γ) and tumor necrosis factor-α (TNF-α) triggers T-cell activation, thereby evoking an adaptive immune response against the xenograft [11,12,13]. While the removal of major xenoantigens from genetically modified pig cells can prevent NK cell-mediated ADCC, the mediators of direct cytotoxicity of these genetically modified pig endothelial cells by human NK cells via receptor–ligand interactions remain largely unknown [14].

NK cells express a diverse array of inhibitory and activating receptors, which engage with their ligands on the target cells to distinguish self, non-self, and abnormal cells [15]. The equilibrium between these inhibitory and activating signals governs the function of NK cells. In xenotransplantation, the inhibitory signals are lacking because MHC class I molecules are not recognized by the inhibitory receptors on NK cells across species [14,16,17]. Therefore, the dominance of activating signals results in the killing of pig endothelial cells by human NK cells.

Human NKG2D has been identified as a pivotal activating receptor in xenogeneic human NK cell-mediated cytotoxicity against pig cells [9,10]. NKG2D ligands include ULBPs and MHC class I chain-related protein (MIC) A and B in humans [18,19], as well as Rae-1 and H-60 in mice [20,21].

To date, the functional role of porcine ligands for the human NKG2D receptor is not yet clear; however, previous studies have established a potential relationship for specific porcine ligands. For instance, porcine *UL-16 binding protein 1 (pULBP-1)* has been cloned and characterized as a homolog of human *ULBP* [22]. The sequence of porcine *MIC2* (*pMIC2*) also confirmed that pMIC2 is a homologue of human MIC proteins [23]. A study by Lilienfeld et al. reported that pULBP-1, but not pMIC2, is the predominant functional ligand for human NKG2D [24]. Contradictory to these findings, Tran et al. revealed that NKG2D-dependent cytotoxicity persisted even after enzymatic removal of pULBP-1 from porcine renal epithelial and intestinal endothelial cells, thereby implicating the presence of other ligands capable of binding to human NKG2D receptor and inducing NK cell activation [25]. Currently, the role of pULBP-1 as the primary trigger for NK cell activation in the context of pig-to-human xenotransplantation has yet to be established.

The present study re-evaluated the involvement of porcine ULBP-1 in human NK cell activation. For this purpose, we employed CRISPR/Cas9 gene editing technology to disrupt the *pULBP-1* gene in a 5-gene knockout porcine endothelial cell line (*GGTA1*, *CMAH*, *β4galNT2*, *SLA-I α chain*, and *β-2 microglobulin*, 5GKO). The 5GKO cells have significantly reduced their immunogenicity by eliminating three major carbohydrate xenoantigens (α Gal, HD, and Sda) as well as the swine leukocyte antigen I (SLA-I α chain and β-2 microglobulin). Those proteins are major xenoantigens for antibody-mediated immune rejection. However, the 5GKO cells retain the capability to activate human NK cells, similar to wild type (WT) pig cells [14]. Subsequently, we investigated the binding capability of pULBP-1-deficient pig cells with a human NKG2D-Fc chimera protein. Additionally, we determined the extent of activation and cytotoxicity exhibited by human NK cells in response to these pULBP-1-deficient pig cells. The findings from this study indicate that pULBP-1 is not a functional ligand for the human NKG2D receptor.

## 2. Materials and Methods

### 2.1. Maintenance of Immortalized Porcine Liver-Derived Endothelial Cells (ipLDEC)

ipLDEC with five-gene knockout (5GKO) were cultured on Biocoat collagen I-coated plates or dishes (Corning Incorporated, Corning, NY, USA) in media (a-MEM:EGM-MV 3:1) (Invitrogen/Lonza, Switzerland) supplemented with 10% FBS (Hyclone, Logan, UT, USA), 10% horse serum (Invitrogen, Carlsbad, CA, USA), 12 mM HEPES, and 1% pen/strep (Invitrogen), as described in our previous studies [14,26].

### 2.2. CRISPR/Cas9-Mediated Disruption of the Porcine ULBP-1 Gene

Two target sequences in exon 2 of the *pULBP-1* gene (ATCCAATCCGTGAGTTCCCTGGG and ATCGAAACACTCAAAGACGTTGG) were selected for guide RNA (gRNA) recognition. For *ULBP-1* gRNA production, two pairs of primers (forward 5′ CACCGATCCAATCCGTGAGTTCCCT 3′/reverse 5′ AAACAGGGAACTCACGGATTGGATC 3′; forward 5′ CACCGTCGAAACACTCAAAGACGT 3′/reverse 5′ AAACACGTCTTTGAGTGTTTCGAC 3′) were synthesized (IDT, Coralville, IA), dimerized, and cloned to the CRISPR/Cas9 vector pSpCas9(BB)-2A-GFP (pX458), a gift from Dr. Feng Zhang [27] (Addgene plasmid # 48138), resulting in the pX458-*pULBP-1A* and pX458-*pULBP-1B* gRNA plasmid constructs. 5GKO cells were co-transfected with pX458-*pULBP-1A&B* gRNAs as previously described [28]. Briefly, 1 µg of each plasmid was added to 1 × 10^6^ of 5GKO cells. The cell mixture was electroporated at 1300 V, 30 ms, 1 pulse using Neon Transfection System (Thermo Fisher Scientific, Waltham, MA, USA). Two days after transfection, GFP-expressing 5GKO cells were enriched by flow sorting. Sorted GFP-positive cells were placed on three 10 cm dishes at low density (500 cells/dish) and cultured for 10 days in a 5% CO_2_ incubator at 37 °C. Twenty single-cell clones were isolated and expended. Cells from each clone were lysed for PCR analysis. A pair of primers to amplify the *pULBP1A&B* gRNAs targeted region was designed and synthesized (IDT): forward 5′ ACTTCATCATCATTCCAAAGCC 3′/reverse 5′ CACTTGGATTTCCTGACTCACC 3′. PCR product was purified and sequenced using a sequencing primer 5′ ACTTCATCATCATTCCAAAGCC 3′ (Azenta Life Sciences, South Plainfield, NJ, USA).

Once mutation was confirmed by Sanger sequencing, PCR fragments were cloned into Zero Blunt TOPO vector (Thermo Fisher Scientific, Waltham, MA, USA). The mutations in both alleles were further revealed by Sanger sequencing using M13F primer. A cell line bearing frame-shift mutations in both *pULBP-1* alleles was selected as 6-gene knockout cell line (6GKO, *GGTA1*/*CMAH*/*β4galNT2*/*SLA-I α chain*/*β-2 microglobulin/pULBP-1*). The mutated pULBP-1 protein sequences were predicted using MacVector software (MacVector, Inc., Apex, NC, USA).

### 2.3. Binding of pULBP-1 Deficient Cells to Human NKG2D-Fc Chimera Protein

pULBP-1-deficient cells (6GKO) were harvested, washed in RPMI 1640 media with 0.1% NaN_3_, and incubated with recombinant human NKG2D-Fc chimera protein (R&D Systems, Minneapolis, MN, USA) at 5 µg/mL for one hour at room temperature. Cells were washed three times in RPMI 1640 media with 0.1% NaN_3_ and stained with Alexa Fluor^®^ 488 AffiniPure F(ab’)_2_ Fragment Donkey Anti-Human IgG, Fcγ fragment specific (Jackson ImmunoResearch Laboratories, West Grove, PA, USA) at 7 μg/mL (1:200) for 45 min at room temperature. Cells were washed and subjected to flow cytometry using an LSRFortessa analyzer (X-20) (BD Bioscience, San Jose, CA, USA). 5GKO cells binding to NKG2D-Fc chimera protein was used as a positive control. 5GKO or 6GKO cells stained with Alexa Fluor^®^ 488 AffiniPure F(ab’)_2_ Fragment Donkey Anti-Human IgG alone were used as a negative control (background). Three independent experiments were performed.

### 2.4. Human NK Cell Activation in Response to Porcine Endothelial Cell Stimulation

NK cell activation was assessed by CD107a degranulation assays as previously reported [29]. Buffy coats were purchased from Versiti Indiana Blood Center. Fresh whole blood was drawn from two volunteers according to the guidelines of the Institutional Review Board (IRB) of Indiana University, IRB#11013. Ficoll-Paque Plus (GE-Healthcare, Pittsburgh, PA, USA) and Lymphoprep (STEMCELL Technologies, Vancouver, Canada) gradient centrifugations were used to isolate human peripheral blood mononuclear cells (PBMCs) from buffy coats and fresh whole blood, respectively. PBMCs were cultured in the RPMI1640 medium supplemented with 10% FBS, 1% penicillin/streptomycin, and 20 ng/mL recombinant human IL-2 (rhIL-2) (BioLegend, San Diego, CA, USA) at 37 °C in a 5% CO_2_ incubator for 5 days. The day before co-culture, 5GKO and 6GKO cells were seeded at 5 × 10^4^ per well in a Biocoat collagen I-coated 48-well plate (Corning Incorporated, Corning, NY, USA). PBMCs treated with rhIL-2 were added to pig endothelial cells at 5 × 10^5^ per well and the cells were co-cultured for 2 h at 37 °C in a CO_2_ incubator. For antibody blocking studies, rhIL-2 treated PBMC were pre-incubated with anti-NKG2D antibody and isotype antibody at 10 µg/mL (BioLegend, San Diego, CA, USA) at 4 °C for 30 min, then co-cultured with pig cells for 2 h at 37 °C in a CO_2_ incubator. Co-cultured cells were collected and stained with fixable viability dye eFluor 780 (Thermo Fisher Scientific, Waltham, MA, USA) to distinguish between live and dead cells, followed by staining with fluorochrome-conjugated antibodies against human CD45, CD3, CD56, and CD107a (BioLegend, San Diego, CA, USA). After fixing with 2% paraformaldehyde (PFA), stained cells were then acquired using an LSRFortessa flow cytometer (BD Biosciences, Franklin Lakes, NJ, USA) and analyzed using FlowJo v10 software (BD Biosciences, Franklin Lakes, NJ, USA). NK cell activation was assessed by the frequency of CD107a-expressing cells within the live singlet CD45^+^ CD3^-^CD56^+^ NK cell population.

### 2.5. Calcein-AM Release Assay

Human NK cell-mediated cytotoxicity of pig endothelial cells was measured by the Calcein-AM assay [30]. Pig endothelial cells were labeled with Calcein AM (BioLegend, San Diego, CA, USA) according to manufacturer’s protocol. The ratios of human PBMC and pig endothelial cells at 50:1, 25:1, and 10:1 were tested in a pilot experiment. A ratio of 10:1 (PBMC:pig endothelial cell) was used in subsequent experiments. A million WT, 5GKO, and 6GKO cells were labeled with 0.01 µM Calcein-AM for 30 min at 37 °C in the dark. Labeled cells were co-cultured with rhIL-2 activated human PBMCs (*n* = 6) in triplicate. After incubation at 37 °C in a 5% CO_2_ incubator for 4 h, 100 μL of each supernatant was harvested and transferred into a 96-well microplate for fluorescence-based assays (Thermo Fisher Scientific, Waltham, MA, USA). Samples were measured using a Cytation 5 imaging reader (BioTek Instruments, Inc., Winooski, VT, USA) at an excitation wavelength of 485 nm and emission wavelength of 530 nm. The fluorescence of culture media alone was subtracted from all samples. Specific lysis was calculated according to the formula [(test release − spontaneous release)/(maximum release − spontaneous release)] × 100. Spontaneous release represents Calcein-AM release from pig cells in medium alone, and maximum release is Calcein-AM release from pig cells lysed in medium with 0.4% Triton X-100. Each sample was assessed in a minimum of three replicate wells, ensuring that the data collected is more reliable and statistically meaningful.

### 2.6. Statistical Analysis

Statistical analyses were performed using GraphPad Prism 9 software (GraphPad Software, San Diego, CA, USA). Data are expressed as mean ± standard error of the mean (SEM). A one-way ANOVA was used to determine statistically significant difference among multiple groups. The difference between two groups was calculated with Student’s *t*-test. A *p*-value less than 0.05 was considered statistically significant. * *p* < 0.05; ** *p* < 0.01; ns: not significant.

## 3. Results

### 3.1. NKG2D Plays a Pivotal Role in Human NK Cell-to-Pig Endothelial Cell Xenogeneic Immune Response

In the present study, we utilized PBMCs to allow for a more physiologically representative NK cell activation to account for interactions with other immune cells. In healthy individuals, NK cells constitute approximately 5–15% of the total PBMC population. NK cell activation often depends on cytokine signals, such as IL-2 and IL-15, and these signals originate from T cells or other immune cells [31]. We assessed human NK cell activation by monitoring CD107a expression within the CD3^-^CD56^+^ subset (NK cells). CD107a is also known as lysosomal-associated membrane protein-1 (LAMP-1), and CD107a serves as a functional indicator for NK cell activation in that CD107a surface expression correlates with cytokine production as well as NK cell-mediated cytotoxicity [32]. In our experiment, we examined PBMCs from three donors (*n* = 3), and we found that anti-NKG2D antibody exhibited strong inhibitory effects on NK cell activation. The presence of anti-NKG2D antibody led to a significant reduction in CD107a expression on NK cells compared to isotype control antibody (*p* < 0.01, 18.87% ± 1.24% vs. 48.13% ± 2.55%) upon WT pig cell stimulation (Figure 1). These findings confirm the pivotal role of NKG2D receptor and its ligands in activating NK cells.

### 3.2. Disruption of pULBP-1 Gene in 5GKO Cells

To disrupt the *pULBP-1* gene in 5GKO cells, 5GKO cells were transfected with two pX458-*pULBP-1* gRNAs targeting different regions in exon 2 of *pULBP-1* gene. Transfected cells were enriched by flow sorting of green fluorescence protein (GFP)-expressing cells. To identify a cell line bearing mutations in both alleles of the *pULBP-1* gene, single sorted cells were expanded and further screened by PCR and Sanger sequence. We ultimately identified a cell line with two allele mutations: one allele with 3 nucleotides deleted and another allele with 61 nucleotides deleted (Figure 2A). Mutations occurred in both gRNA-targeted regions. These mutations resulted in frameshifts, and predictive software indicated that the two mutated alleles encoded truncated pULBP-1 proteins with 72 and 79 amino acids, respectively (Figure 2B). Because these mutations led to the loss of glycosylphosphatidylinositol (GPI) anchor signal in pULBP-1, the truncated pULBP-1 proteins are no longer retained in the cell membrane.

### 3.3. pULBP-1-Deficient Cells Exhibit a Reduced Binding Potential to NKG2D-Fc Chimera Protein

The interaction of pULBP-1 ligand and human NKG2D receptor has been previously reported [24,33]. To determine whether the binding of NKG2D to pULBP-1-deficient porcine cells was altered, 6GKO and 5GKO cells were incubated with a recombinant human NKG2D-Fc chimera protein, followed by the staining with Alexa Fluor^®^ 488 AffiniPure F(ab’)_2_ Fragment Donkey Anti-Human IgG, Fcγ fragment-specific. A representative flow cytometry result showed that human NKG2D-Fc binding to 6GKO cells was reduced compared to the binding to 5GKO cells (Figure 3A). The mean fluorescent intensity (MFI) of NKG2D-Fc binding was 7527 ± 1081 for 6GKO and 12,729 ± 1331 for 5GKO (presented as mean ± SEM). 6GKO cells exhibited 40.9% reduction in binding to NKG2D-Fc protein compared to the parental 5GKO cells (*p* < 0.05, *n* = 3) (Figure 3B). This result demonstrated the absence of the pULBP-1 protein on 6GKO cells as well as the presence of unidentified ligands on pig cells that can bind to the human NKG2D-Fc protein.

### 3.4. pULBP-1 Does Not Contribute to Xenogeneic Human NK Cell Activation

Because previous studies examining the role of pULBP-1 in human NK cell activation led to contradictory findings [24,25], we directly compared human NK cell activation in response to the stimulation of 5GKO and 6GKO cells. Our previous study showed that 5GKO cells induced human NK cell activation similar to WT and TKO (triple-gene knockout, *GGTA1/CMAH/β4galNT2*) cells [14]. The flow cytometry gating strategy to identify live human CD3^-^CD56^+^ NK cells is shown in Figure 4A. Representative flow plots showing human NK cell degranulation upon the stimulation of 5GKO or 6GKO cells are shown in Figure 4B. Human NK cell activation was measured by the percentage of CD107a positive cells in NK cell population. Student’s t-test indicated no significant difference between 5GKO and 6GKO cells in stimulating human NK cell degranulation (*n* = 8, *p* = 0.4962).

### 3.5. pULBP-1 on Porcine Cells Has No Effect on Human NK Cell-Mediated Cytotoxicity

Human NK cell cytotoxicity was measured by the Calcein-AM assay [30]. A pilot study was conducted to test the ratio of human PBMCs and porcine endothelial cells. The killing activity was at 70.3%, 56.5%, and 52.7%, respectively, with human PBMCs to pig cells at ratios of 50:1, 25:1, and 10:1 (Figure 5A). Therefore, based on these findings, a ratio of 10:1 was selected for subsequent experiments. The result indicated no statistically significant difference in NK cell cytotoxicity between human NK cells stimulated by 5GKO and 6GKO porcine cells (*p* = 0.9941). Therefore, the pULBP-1 on porcine cells is not a functional ligand for human NK cell-mediated cytotoxicity (Figure 5B). WT porcine cells were also tested in this experiment, and there was no significant difference in NK cell cytotoxicity between human NK cells stimulated by WT and 6GKO porcine cells (*p* = 0.2946).

## 4. Discussion

The goal of this study was to re-evaluate the involvement of porcine ULBP-1 in human NK cells activation and its role as a primary trigger for NK cell activation in the context of pig-to-human xenotransplantation. Using CRISPR/Cas9-mediated gene editing, we assessed whether the pULBP-1 is a potential NKG2D ligand. We successfully eliminated the pULBP-1 protein from pig endothelial cells. The wildtype *pULBP-1* gene encodes 247 amino acids. We introduced the mutations in both alleles of the *pULBP-1* gene, thereby resulting in the production of truncated ULBP-1 proteins encoding 72 and 79 amino acids. These truncated pULBP-1 were no longer retained in the cell membrane due to the lack of the GPI anchor [18,25]. In addition, we found that pULBP-1-deficient pig cells showed approximately a 40.9% reduction in binding capacity to human NKG2D-Fc compared to the control cells. These findings are consistent with the absence of ULBP-1 protein. Next, we compared human NK cell activation stimulated by pULBP-1-deficient pig cells as well as control pig cells. Surprisingly, despite a significant reduction in binding, we observed no significant difference in NK cell degranulation or cytotoxicity. Therefore, these findings suggest that pULBP-1 is not a functional ligand for the human NKG2D receptor.

These findings from the present study are inconsistent with a previous report suggesting that pULBP-1 is a critical ligand for human NKG2D receptor, which leads to NK cell-mediated cytotoxicity [24]. For instance, by reducing pULBP-1 mRNA expression or introducing anti-pULBP-1 polyclonal antibody, Lilienfeld et al. found that human NK cell-mediated cytotoxicity was partially reduced or completely blocked in response to pig endothelial cells. In addition, a *pULBP-1* knockout pig was previously generated, and the findings indicated that porcine aortic endothelial cells deficient in pULBP-1 exhibited higher cell viability (85.36%) compared to the control group (69.58%) in a cytotoxicity assay using a human NK cell line NK92MI (*p* = 0.0565) [34].

To shed further light on the role of pULBP-1 in initiating NK cell activation, we used pULBP-1-deficient endothelial cells to stimulate human NK cell responses. We observed reduced, but significant NKG2D-Fc binding on pULBP-1-deficient pig cells, which indicated the presence of unidentified NKG2D ligands. This observation is consistent with previous research [25]. Indeed, these unidentified ligands on pig cells likely interact with the NKG2D receptor to initiate NK cell activation, thereby leading to the destruction of pig cells. Identifying these unknown porcine NKG2D ligands warrants future studies, as those ligands can potentially be used to manipulate NKG2D signals and block human NK cell activation in xenotransplantation.

An alternative strategy to regulate xenoreactive human NK cells is to introduce HLA class I molecules into pig cells, providing inhibitory signals to mitigate NK cell activation. Pig cells are susceptible to destruction by human NK cells due to imbalanced inhibitory and activating signals. Unlike human leukocyte antigen (HLA) class I molecules, swine leukocyte antigen class I (SLA-I) molecules cannot be recognized by inhibitory receptors on human NK cells [14,16]. Therefore, the dominance of activating signals initiates human NK cell activation. Several studies have shown that porcine cells expressing non-classical HLA class I molecules, including HLA-E and/or HLA-G, can protect porcine cells from human NK cell cytotoxicity [35,36,37]. Further, our recent study indicated that co-expressing HLA-E and HLA-G on porcine cells effectively inhibits human NK cell activation [29].

## 5. Conclusions

Our study confirmed that the human NK cell activating receptor NKG2D played a crucial role in the human NK cell-to-pig endothelial cell xenogeneic immune response. We further demonstrated that pULBP-1 could bind to the NKG2D chimera protein, but it was dispensable for NKG2D-mediated NK cell activation. Future studies will focus on identifying and eliminating porcine ligands for human NKG2D receptor from porcine cells, as well as introducing HLA-E and HLA-G inhibitory ligands into porcine cells through genetic engineering. These strategies have great potential to block human NK cell activation and NK cell-mediated rejection [38]. Additionally, investigating the interplay between NK cells and other immune cell populations, such as macrophages and T cells, could provide a more comprehensive understanding of the immune response to xenogeneic cells. Together, our study underscores the necessity for comprehensive investigations into the ligand–receptor interactions involved in NK cell activation and emphasizes the importance of considering the broader context of immune recognition in xenotransplantation.

## Figures and Tables

**Figure 1 cells-12-02587-f001:**
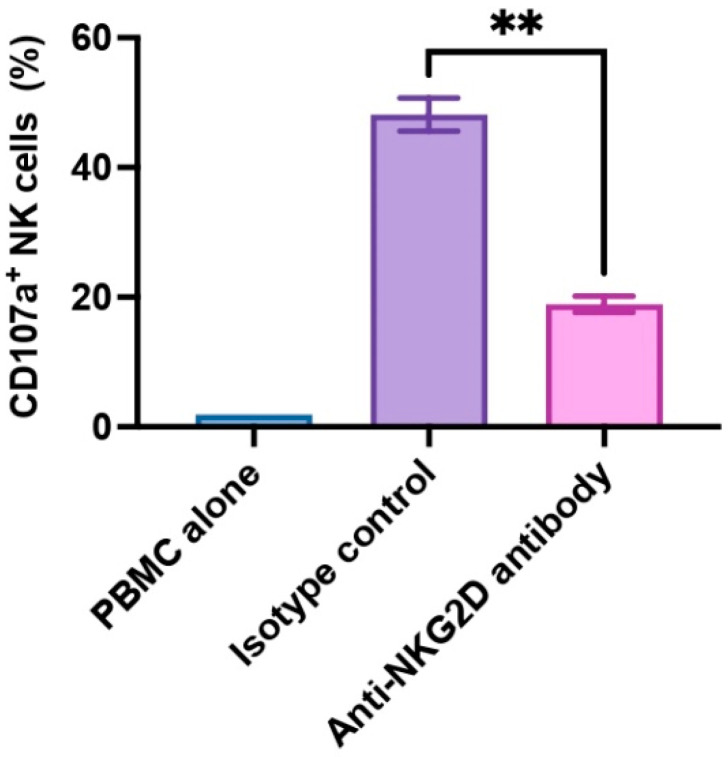
NKG2D receptor-mediated xenogeneic human NK cell activation. Human PBMCs (*n* = 3) were treated with rhIL-2 for 5 days, then pre-incubated with anti-human NKG2D antibody or an isotype control antibody at 10 µg/mL for 30 min, and co-cultured with WT pig endothelial cells for 2 h. Human NK cell activation was assessed by the frequency of CD107a-expressing cells in the CD3^−^CD56^+^ NK cell population. PBMC alone was used as an unstimulated control. The isotype antibody-treated group was used as a control. Data were shown as mean ± SEM. Paired t-test was performed to analyze the differences between the two groups. ** *p* < 0.01.

**Figure 2 cells-12-02587-f002:**
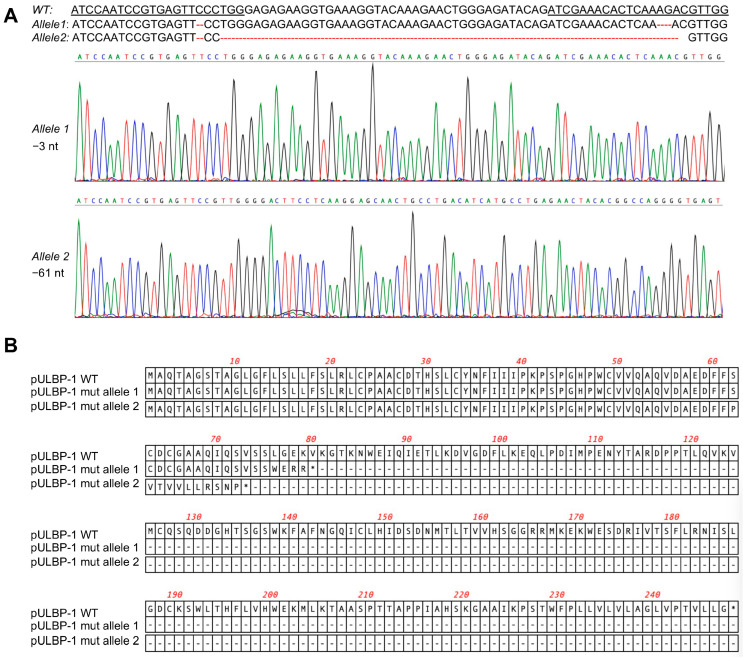
Elimination of pULBP-1 in pig endothelial cells. (**A**) Disruption of *pULBP-1 gene*. CRISPR/Cas9 gRNAs targeted *pULBP-1* DNA sequences were underlined. The missing nucleotides were marked using dash line in red. Sanger DNA sequencing chromatograms of the two *pULBP-1* mutant alleles in 6GKO were shown. Three nucleotides were deleted in *allele 1*, and sixty-one nucleotides were missing in *allele 2*. (**B**) Translation of mutated *pULBP-1*. Deletion of 3 nucleotides in *allele 1* led to a truncated pULBP-1 with 79 amino acids. Deletion of 61 nucleotides in *allele 2* led to a truncated pULBP-1 with 72 amino acids.

**Figure 3 cells-12-02587-f003:**
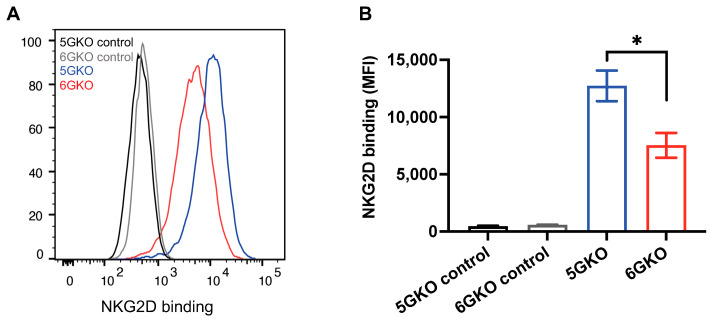
pULBP-1 deficient cells exhibited a reduced binding capability to human NKG2D-Fc. (**A**) Representative binding results of NKG2D-Fc to pig cells. pULBP-1 deficient 5GKO cells (6GKO) were incubated with recombinant human NKG2D-Fc at 5 µg/mL followed by staining with Alexa Fluor 488-AffiniPure F(ab’)2 Fragment Donkey anti-Human IgG shown in red. 5GKO cells binding to human NKG2D-Fc was used as a positive control shown in blue. 5GKO and 6GKO cells stained with Alexa Fluor 488-AffiniPure F(ab’)2 Fragment Donkey anti-Human IgG alone were used as negative controls, shown in black and grey, respectively. (**B**) 6GKO cell binding to NKG2D-Fc chimera protein was significantly reduced compared to 5GKO cell binding to NKG2D-Fc chimera protein. Data were shown as mean ± SEM. Student’s t-test was performed to analyze the difference between the two groups. * *p* < 0.05.

**Figure 4 cells-12-02587-f004:**
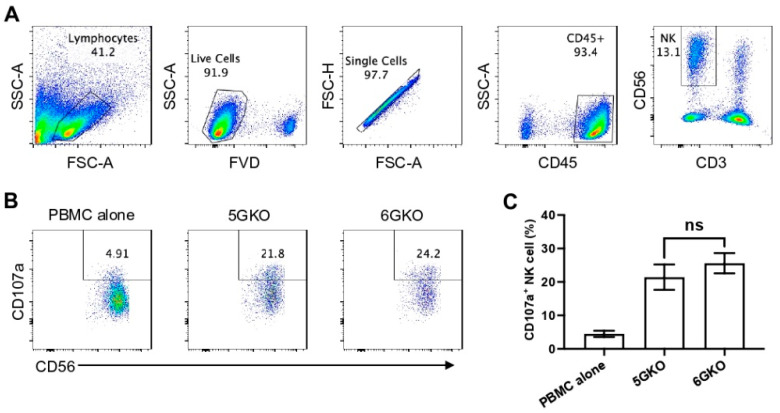
Human NK cell activation in response to the stimulation of pULBP-1-deficient pig cells. 5GKO and 6GKO cells were co-cultured with rhIL-2 treated PBMCs (n = 8) for 2 h. (**A**) The flow cytometry gating strategies to show the live NK cell population. (**B**) Representative flow plots of NK cell activation to the modified porcine cells simulation assessed by the percentage of CD107a positive cells in the CD3^-^CD56^+^ population. PBMC alone was used as a control. (**C**) pULBP-1 deficiency had no impact on human NK cell degranulation. Data presented as mean ± SEM. The differences between the two groups were analyzed by Student’s *t*-test. ns, not significant.

**Figure 5 cells-12-02587-f005:**
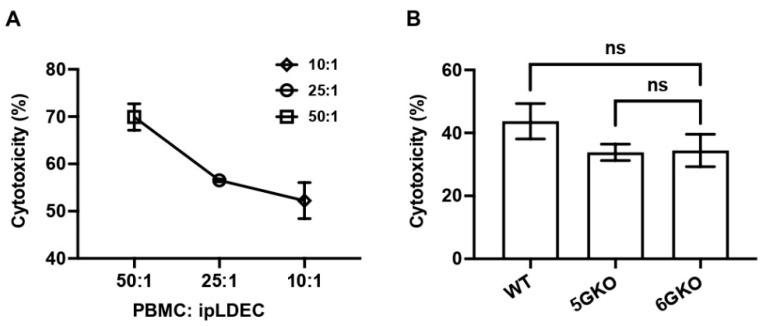
pULBP-1 is dispensable for pig endothelial cells triggered human NK cell cytotoxicity. (**A**) The ratios of human PBMCs to pig endothelial cells at 50:1, 25:1, and 10:1 were tested in the Calcein-AM assay. A ratio of 10:1 was chosen in subsequent cytotoxicity assays. (**B**) Human PBMCs (n = 6) were co-cultured with Calcein-AM labeled WT, 5GKO, and 6GKO cells for 4 h. Fluorescence of the co-culture supernatant was measured by a Cytation 5 imaging reader. Cytotoxicity was calculated as described in Materials and Methods. Data were presented as mean ± SEM. One-way ANOVA was used to analyze the differences among the groups. ns: not significant.

## Data Availability

Sus scrofa UL16 binding protein 1 (ULBP1) mRNA, NCBI reference sequence: NM_001004035.1.

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
