# Peer review of "Porcine UL-16 Binding Protein 1 Is Not a Functional Ligand for the Human Natural Killer Cell Activating Receptor NKG2D"

_cells, 2023, doi:10.3390/cells12222587_

Round 1
Reviewer 1 Report
Comments and Suggestions for Authors
Heterologous organ transplantation provides an effective solution to the severe shortage of donor organs in human organ transplantation. The latest progress in pig genetic engineering has facilitated the use of heterologous donor organs in human clinical settings. However, immunological rejection remains a major obstacle to improving the survival rate of pig-to-human xenotransplantation. Natural killer (NK) cells play a crucial role in the rejection response during heterologous transplantation. One approach to inducing NK cell immune tolerance is to block the direct killing of pig cells by NK cells through the interaction between the activating receptor NKG2D and its ligands.
Kevin J. Lopez et al. reevaluated the involvement of pig ULBP-1 in the activation of human NK cells in human-pig xenotransplantation and its role as a major triggering factor for NK cell activation. The study found that pig cells lacking pULBP-1 had reduced binding capacity to human NKG2D-Fc and did not significantly affect the activation and cytotoxicity of stimulated human NK cells. Therefore, these results demonstrate that pULBP-1 is not the key ligand for activating human NK cells in the process of xenotransplantation.
This study provides new insights into the ligand-receptor interactions involving pig NKG2D ligands, shedding light on unknown mechanisms involved in NK cell activation during xenotransplantation. Therefore, this article should be recommended for publication in CELLS.
The author needs to address the following questions:
1. Why was pULBP-1 selected for knockout based on five-gene modification?
2. What is the targeting efficiency of 6GKO knockout in pig cell lines?
3. For gene editing methods utilizing the CRISPR/Cas9 system, I would like the authors to examine the potential off-target sites of this method.
Author Response
Thank you for your comments, please see the attached responses.

Reviewer 2 Report
Comments and Suggestions for Authors
A brief summary:
This research article investigated whether porcine UL-16 binding protein 1 (pULBP-1) on porcine endothelial cells could act as a ligand of NKG2D of NK cells to mediate the cytotoxic effects of NK cells on xenogeneic cells. The authors employed CRISPR/Cas9 gene editing technology to disrupt the ULBP-1 gene in a genetically modified pig cell line and investigated the NKG2D binding capability of pULBP-1-deficient pig cells as well as activation and cytotoxicity of human NK cells reacted to pULBP-1-deficient pig cells. However, the results showed that the loss of pULBP-1 on cell surface didn’t affect the activation of NK cells by xenogeneic cells, indicating that pULBP-1 is not a functional ligand for the human NKG2D receptor as its human ULBP homolog. This finding could help to find future therapies that aim to block NKG2D signaling to prevent xenotransplantation rejection.
.
Specific comments:
1. Unlike T cells, NK cells do not recognize antigens, instead, they distinguish non-self cells from self cells through the function of an array of activating and inhibitory receptors that recognize self-proteins expressed on the cellular surface. Although NKG2D is a key activating receptor, others such as DNAX accessory molecule 1 (DNAM-1), and the natural cytotoxicity receptors natural killer P30, 44, and 46-related protein (NKp30, NKp44, and NKp46) could also play a role in activating NK cells. In addition, the lack of blocking on inhibitory receptors like inhibitory killer cell immunoglobulin-like receptors (KIRs) and the heterodimer CD94-natural killer group 2A (NKG2A) could also let NK cells in activating state. Thus pULBP-1 could still play a role in activating NK cells, but due to multiple activation pathways, the loss of pULBP-1 could still not turn down the ability of xenogeneic cell to activate NK cells since some pathway may compensate it. Other activating singals or inhibitory signals should be considered in this study to investigate role of pULBP-1 on NK cell activation.
2. To comprehensively study the role of pULBP-1 on NK cell activation, WT cells and pULBP-1 single KO cells along with 5GKO and 6GKO cells all should be included in experiments. Only Figure 5 included WT cell controls and the data showed that 5GKO and 6GKO numerically reduced NK cell cytotoxicity. And for the same justification, since NK cells only constitute 5% to 15% of the total PBMC population, including a human NK cell line like NK92 in all NK cell related experiments could also increase the robustness of this study.
3. In Figure. 1. pig endothelial cells were mentioned to be used. Please clarify which type of pig endothelial cells.
4. The experiments demonstrate the loss of pULBP-1 on cell surface of pULBP-1-deficient cells could strength the evidences despite that the authors mentioned that “Because these mutations led to the loss of glycerylphospho inositol (GPI) anchor signal in pULBP-1, the truncated pULBP-1 proteins are no longer retained in the cell membrane”. Additionally, pULBP-1 overexpression approach is suggested to be used to clarify the role of pULBP-1 on NK cell activation.
Author Response

(The authors gave the same response as above.)

Round 2
Reviewer 2 Report
Comments and Suggestions for Authors
The authors have addressed most of my concerns.
Author Response
Thank you.